# INTERACTIONS BETWEEN REPRESENTATION LEARNING AND SUPERVISION

## ABSTRACT

Representation learning is one of the fundamental problems of machine learning. On its own, this problem can be cast as an unsupervised dimensionality reduction problem. However, representation learning is often also used as an implicit step in supervised learning (SL) or reinforcement learning (RL) problems. In this paper, we study the possible "interference" supervision, commonly provided through a loss function in SL or a reward function in RL, might have on learning representations, through the lens of learning from limited data and continual learning. Particularly, in connectionist networks, we often face the problem of catastrophic interference whereby changes in the data distribution cause networks to fail to remember previously learned information (French, 1999) and learning representations can be done without labeled data. A primary running hypothesis is that representations learned using unsupervised learning are more robust to changes in the data distribution as compared to the intermediate representations learned when using supervision because supervision interferes with otherwise "unconstrained" representation learning objectives. To empirically test hypotheses, we perform experiments using a standard dataset for continual learning, permuted MNIST (Goodfellow et al., 2013; Kirkpatrick et al., 2017; Zenke et al., 2017). Additionally, through a heuristic quantifying the amount of change in the data distribution, we verify that the results are statistically significant.

## 1 INTRODUCTION

Consider a taxonomy by which machine learning algorithms are traditionally differentiated, i.e., based on whether they fall under the supervised, unsupervised or reinforcement learning paradigms. The major difference between the three is how the objective function is specified, and particularly distinguishing between unsupervised learning and the other two, whether each input has a predefined associated target or not.

Learning a representation on its own can be viewed as an unsupervised learning problem because there is no need for any associated labels or rewards. Specifically, representation learning can be viewed as a dimensionality reduction problem. While it is not necessary to do any processing on data when features are small in number and easy to learn from, it becomes more important as the dimensionality increases and direct feature "usefulness" decreases (the curse of dimensionality.) However, by Occam's razor (Tornay, 1938), it is not unexpected that we have gravitated more to the supervised settings, and focused less on the problem of solely learning "general" representations as it is quite hard and is not really necessary if the goal is to perform well on particular well-defined function approximation style problems.

The main theme behind this study is the general notion that the problem of learning representations of data is separate from or orthogonal to the problem of learning models for fitting functions or making predictions and hence we can consider them separately. Hence, we can ask "Do we lose on the ability to learn better, more general representations by implicitly performing representation learning in supervised settings?"

Learning from limited data is desirable under many scenarios. One such scenario is multi-task learning where multiple related tasks are presented to an agent and its goal is to do well in all of them and generalize to similar tasks. However, this has been a challenging problem due to the difficulty in finding good representations and overcoming catastrophic interference issues. If we

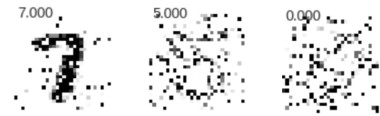

Figure 1: Examples of Permuted MNIST with $\delta = 196(\text{left}), 500(\text{middle}), 784(\text{right})$

solve the representation learning problem well without supervision, we can make best use of data and moreover learn to generalize over multiple tasks better when we have limited labeled data. This is another primary motivation for studying this problem.

## 2 PROPOSED APPROACH

A common architecture for unsupervised learning in Deep Learning is the autoencoder which aims to approximate the identity function by reconstructing inputs from themselves through a bottleneck encoding. Formally, we can describe the autoencoder as a composition of two networks, an encoder network that produces a low-dimensional embedding of the input and a decoder network that attempts to reconstruct the input given the embedding: $\phi_{\mathbf{x}} = \text{Enc}(\mathbf{x}), \tilde{\mathbf{x}} = \text{Dec}(\phi_{\mathbf{x}}) = \text{Dec}(\text{Enc}(\mathbf{x}))$. A common loss function used is the reconstruction error, say mean squared error between the input and reconstruction: $\mathcal{L}(\mathbf{x}, \tilde{\mathbf{x}}) = \text{MSE}(\mathbf{x}, \tilde{\mathbf{x}})$.

Now, let us consider the representation learned when using a "supervision based" objective. Particularly, the representation learned in supervised learning can be described by defining the topology of a sequential neural network more concretely. Suppose a neural network can be written as a composition of $L$ layers: $f = f^{(L)} \circ f^{(L-1)} \circ ... \circ f^{(1)}(\mathbf{x})$ or equivalently, denoting the parameters clearly, $f = f^{(L)}(...f^{(2)}(f^{(1)}(\mathbf{x}; \theta^{(1)}); \theta^{(2)})...; \theta^{(L)})$. Then, we can define the representation learned in the $k^{\text{th}}$ layer as $\psi_{\mathbf{x}}^{(k)} = f^{(k)} \circ ... \circ f^{(1)}$. Given that the last layer is usually where the classification or final output prediction happens, we can then consider comparing $\phi_x$ and $\psi_{\mathbf{x}}^{(L-1)}$. In other words, we can assume the penultimate layer in a supervised learning problem serves as the representation learned and is an equivalent to the encoding produced by an autoencoder.

Finally, as a baseline, we can initialize a network with random weights, as we would before training the supervised agent above, and only train with the last layer on the task. This serves as a test for how good a random representation is.

## 3 EMPIRICAL EVALUATION

A commonly used dataset for continual is Permuted MNIST. As its name suggests, it is based on the the MNIST dataset LeCun & Cortes (2010), a commonly used benchmark dataset for classification comprising of $28 \times 28$ single channel (grayscale) images of handwritten digits, with each image having a corresponding target class from 0 through 9. The original training set consists of 60000 samples while the test set consists of 10000 samples. For permuted MNIST however, we consider defining new "tasks" as corresponding to some random permutation of the pixels. For example, we can create a new task by swapping the first and last pixel of every image. If we do this for both the training and test sets, we get a new task with, again, 60000 training instances and 10000 test instances.

Now, we can define the network architectures. For the autoencoder, we use a symmetrical convolutional autoencoder with 3 layers each for the encoder and decoder. The classifier architecture is similar to the encoder architecture of the autoencoder except that a Dense layer and SoftMax Activation follow the bottleneck. The appendix describes the full architectures.

For the first experiment, we look to measure the robustness of the representations learned using the various approaches. To this end, we consider training the networks on the original MNIST dataset, i.e., without any permutation. For the unsupervised learning setting, this involves learning a representation first and then learning just a final layer for the classification. Similarly, for the baseline, we fix a random representation and learn just a final layer for classification. Next, for all the approaches, we freeze all the layers up to and including the penultimate layer and continue

training on a permuted MNIST task with just the last layer changing. We record the learning curve on the new task while keeping track of the performance on the "test" set, i.e., the originally pre-trained unpermuted MNIST task. We are looking to observe two main metrics: speed of learning and overall accuracy. To observe both of these, we can plot a curve of the average accuracy at fixed intervals for each algorithm for each permutation. To account for the fact that some permutations are "better" or "worse" than others in terms of "distance" from the original unpermuted dataset, we can consider plotting the metrics for different classes of tasks, say $\mathcal{D}_\delta$, where $\delta$ defines the number of swaps made to get the permutation. In other words, given that the original dataset is defined as the permutation $(1, 2, ..., 784)$ and suppose we define a new task using the permutation $(784, 2, 3, ..., 782, 783, 1)$. Then, $\delta = 1$ as the minimum number of pairwise swaps that need to be made to sort this array is 1. In Figure 1, some examples of generated images upon applying permutations with different $\delta's$ are given. Given we generate a number of tasks belonging to classes, say, $\mathcal{D}_{\delta_{\min}}, ..., \mathcal{D}_{\delta_{\max}}$, where reasonable values may be to pick $\delta$'s from a logarithmic scale, we can plot the average accuracy at fixed intervals in time over training, considering all the permutations in a class, i.e., those with the same distance heuristic metric $\delta_i$. If an algorithm does well on the test set for high $\delta$, say it achieves a high threshold level of accuracy quickly or retains a high level of accuracy, it is in some sense robust to large changes in the data distribution and hence we can likely conclude that the algorithm has learned a more general representation that can result better for continual learning and perhaps can withstand adversarial attacks to some extent also. To prove statistical significance, given the same classes of tasks, we can perform paired t-tests, averaging over time, to see whether there is any significant difference in performance between the approaches.

For the second experiment, we follow the traditional continual learning approach whereby the entire network is allowed to change (albeit with different final layers for each task.) However, we additionally modify the experiment slightly to incorporate the heuristic on distance discussed earlier. So, we initially train on, say, the original MNIST dataset. Then, we successively train on tasks with higher and higher $\delta$ values (the data distribution moves further and further away in terms of the minimum number of swaps to get to the initial permutation of $(1, 2, ..., 784)$). Meanwhile, after training on each new dataset, we consider two metrics that are described in French (1999), the exact recognition measure, which defines how worse off we are on previously trained tasks and the savings measure, the amount of time required to relearn on the original data after training on a new task to achieve some level of performance. To make this experiment simple, we again only consider the the initial unpermuted MNIST task when looking at the time to relearn and decrease in performance.

## 4  RESULTS

We can first discuss the results of the first experiment. Recall that in this experiment, we first train the networks on the original unpermuted MNIST dataset. Then, we train the network, fixing the initialized or learned representation (i.e., just fine-tuning the last layer) on a permuted MNIST task. We can then compare the loss in performance on a "test" set, the previously trained unpermuted MNIST task.

In Figure 2, the average classification accuracy for the algorithms for different values of $\delta$, and hence different amounts of change in the data distribution can be seen. After initial experimentation, we use a logarithmic scale starting from $\delta = 400$ for all the experiments. This is because when we used $\delta$ smaller than 400, the classifier and autoencoder both do quite well as the difference in the data distribution is possibly too small or that the convolutions are able to capture some regional features robust to noise. As can be seen in the figure however, the classifier seems to do better when $\delta$ is small (see figures in the Appendix for plots with more values of $\delta$). However, as $\delta$ increases, the autoencoder does better and seems to have a lot lower variance. Looking at the training set and test set plots in the Appendix, the baseline seems to improve mostly due to the training set.

On the second experiment, we also see some interesting results. Note first that the classifier undoubtedly does much better on the initial training than the autoencoder. However, as we add tasks, it seems that the accuracy while retraining reduces and actually stays near constant over retraining, eventually performing worse that the autoencoder accuracy while retraining on average. In addition, the autoencoder performance seems to increase and asymptote towards the initial training performance regardless of the number of tasks.

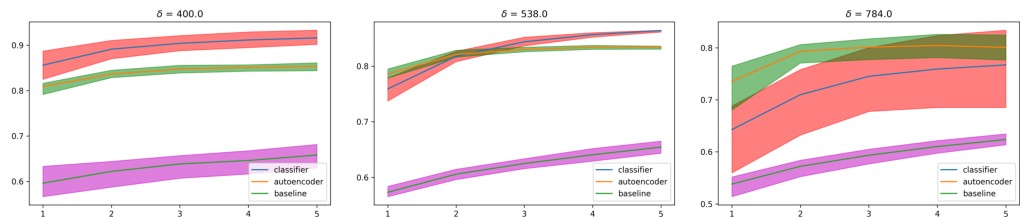

Figure 2: Average Classification Accuracy on Training and Test Sets vs. Epochs

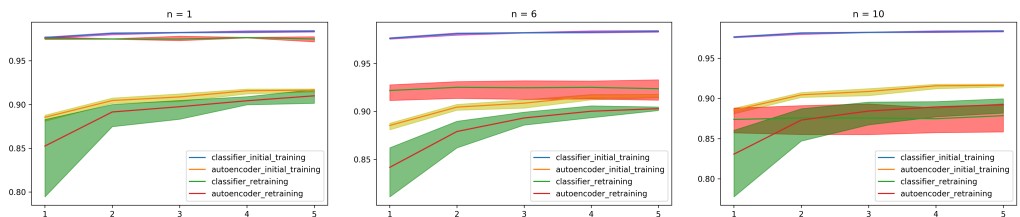

Figure 3: Classification Accuracy on unpermuted MNIST during initial training and retraining after training on $n \in \{1, ..., 10\}$ other permuted MNIST tasks with successively increasing $\delta$'s vs. Epochs

## 5   DISCUSSION

Having briefly noted some important points about the experimental results, we can now discuss what the results mean or perhaps what we can infer from them.

Firstly, we see that the classifier clearly achieves better initial training performance (i.e., before any of the additional experiments that take place to measure robustness, generalization or catastrophic forgetting.) When the whole network is "unconstrained" and the weights are allowed to change, as the case with the supervised setting where we train the network on the unpermuted MNIST allowing all weights to change, we can get much better performance. However, in the unsupervised case, we first train the representation using an autoencoder and then fix the representation while learning a single layer of weights. This constrains or regularizes our problem to some extent. So, we can view this as some sort of bias-variance tradeoff. While the supervised learning setting has lower bias with respect to the original unpermuted MNIST task, it has high variance. This is, for example, clearly seen through the large shaded regions in the test-set classification accuracy graphs in the Appendix when $\delta$ is high for the supervised learner. On the other hand, since the autoencoder is constrained to learn a more general representation, it suffers some loss in performance on the unpermuted MNIST task, but achieves better performance on other tasks.

Another interesting observation is the leveling out or constancy effect described in the previous section. It seems that there is some strong "coupling" effects going on between the representation and final classification layer in the supervised learning case. This suggests that the representation is not general enough to be used across tasks. On the other hand, it is quite intriguing how the autoencoder, even after being trained with 10 other tasks, has a general enough representation that allows a final classification layer to easily fine-tune and achieve near the original accuracy during retraining.

With some of the general observation noted, we conclude this section by noting that statistical significance tests (see Table in the Appendix) also conclude the same results that using unsupervised learning is probably a good idea when continual learning or equivalently robustness or generalizability is desired. Particularly, we can see that with larger changes in the data distribution number or more tasks being added, we accept the null hypothesis more; that the classifier average accuracy is less than or equal to the autoencoder average accuracy (over training or evaluation). That said, it should be noted that it is likely more runs (all the results are given with 3 runs) or some theoretical backing could help really reinforce the arguments made above.

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

## APPENDIX

```
Layer (type)                    Output Shape              Param #
=================================================================
input (InputLayer)              (None, 28, 28, 1)         0

conv_1 (Conv2D)                 (None, 7, 7, 64)          640

conv_2 (Conv2D)                 (None, 3, 3, 128)         73856

conv_3 (Conv2D)                 (None, 1, 1, 256)         295168

flatten (Flatten)               (None, 256)               0

bottleneck (Activation)         (None, 256)               0

pre_activation_output (Dense    (None, 10)                2570

output (Activation)             (None, 10)                0
=================================================================
Total params: 372,234
Trainable params: 372,234
Non-trainable params: 0
```

Figure 4: Classifier Architecture

```
Layer (type)                    Output Shape            Param #
=================================================================
input (InputLayer)              (None, 28, 28, 1)       0

conv_1 (Conv2D)                 (None, 7, 7, 64)        640

conv_2 (Conv2D)                 (None, 3, 3, 128)       73856

conv_3 (Conv2D)                 (None, 1, 1, 256)       295168

flatten (Flatten)               (None, 256)             0

bottleneck (Activation)         (None, 256)             0

reshape_bottleneck (Reshape)    (None, 1, 1, 256)       0

deconv_3 (Conv2DTranspose)      (None, 3, 3, 128)       295040

deconv_2 (Conv2DTranspose)      (None, 7, 7, 64)        73792

deconv_1 (Conv2DTranspose)      (None, 28, 28, 1)       577
=================================================================
Total params: 739,073
Trainable params: 739,073
Non-trainable params: 0
```

Figure 5: Autoencoder Architecture

| $n$ | $t$-Statistic | $p$-Value | $H_0$ | $\delta$ | $t$-Statistic | $p$-Value | $H_0$ |
|-----|---------------|-----------|-------|----------|---------------|-----------|-------|
| 1   | 8.28064       | 0.00058   | ✗     | 400      | 21.137        | 1.5e-5    | ✗     |
| 2   | 8.12894       | 0.00062   | ✗     | 430      | 3.3717        | 0.0139    | ✗     |
| 3   | 7.22272       | 0.00097   | ✗     | 464      | 1.4585        | 0.1092    | ✗     |
| 4   | 6.78498       | 0.00123   | ✗     | 500      | 6.4186        | 0.0015    | ✗     |
| 5   | 6.11040       | 0.00182   | ✗     | 538      | 0.7044        | 0.0018    | ✗     |
| 6   | 3.78353       | 0.00969   | ✗     | 580      | -5.763        | 0.0022    | ✓     |
| 7   | 2.33176       | 0.04005   | ✗     | 626      | -7.439        | 0.0009    | ✓     |
| 8   | 1.14573       | 0.15790   | ✓     | 674      | -3.624        | 0.0111    | ✓     |
| 9   | 0.32978       | 0.37906   | ✓     | 726      | -3.609        | 0.0113    | ✓     |
| 10  | 0.16492       | 0.43850   | ✓     | 784      | -5.543        | 0.0026    | ✓     |

Table 1: Paired t-Test, $H_0$: Supervised representation avg. accuracy $\leq$ Unsupervised representation avg. accuracy, $(p \leq 0.05) \wedge (t > 0)$?

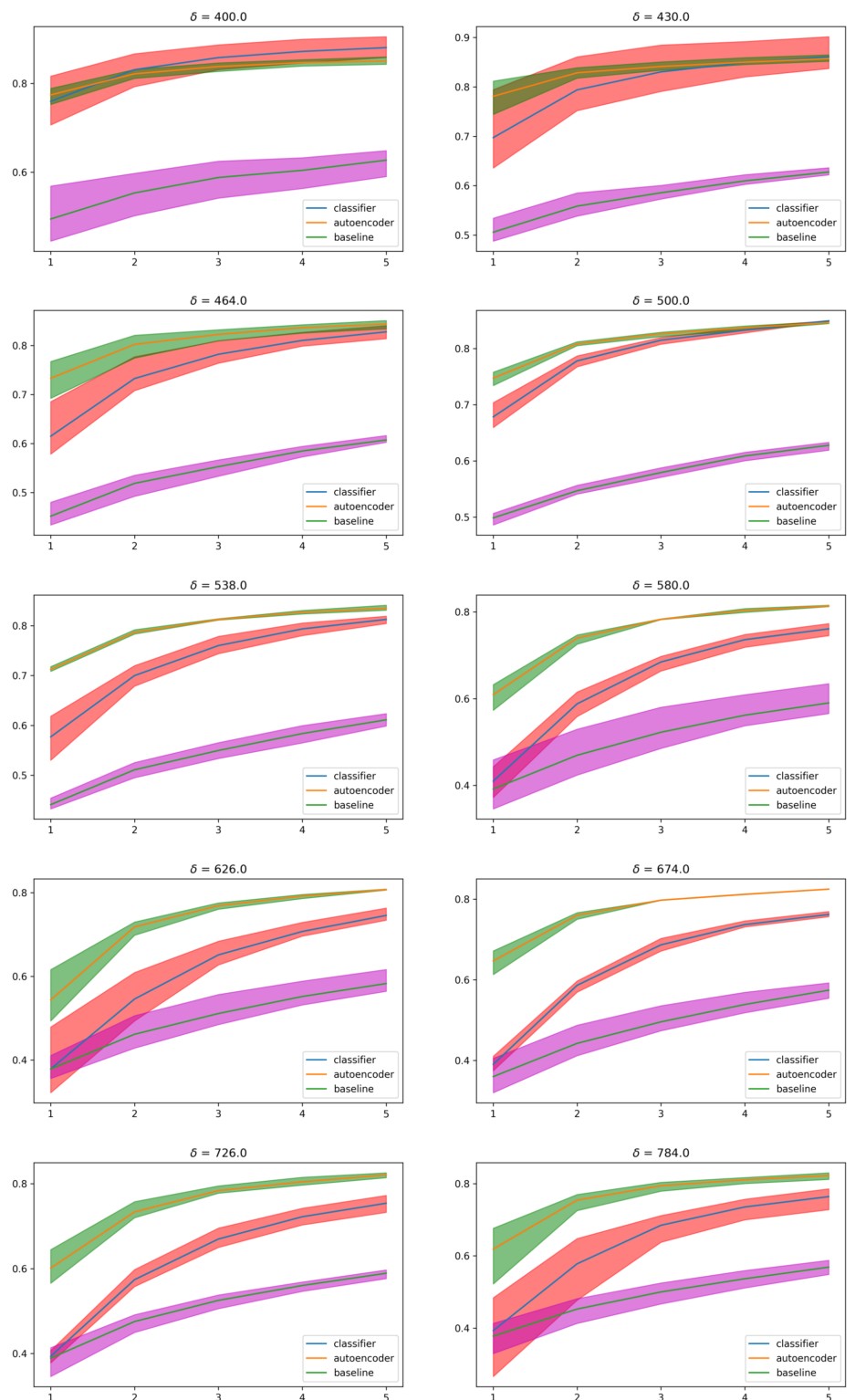

Figure 6: Classification Accuracy on Training Set, i.e., permuted MNIST with distance = $\delta$ vs. Epochs

[1 Epoch = 1024 (Batch Size) $\times$ 100 (Steps Per Epoch) samples]

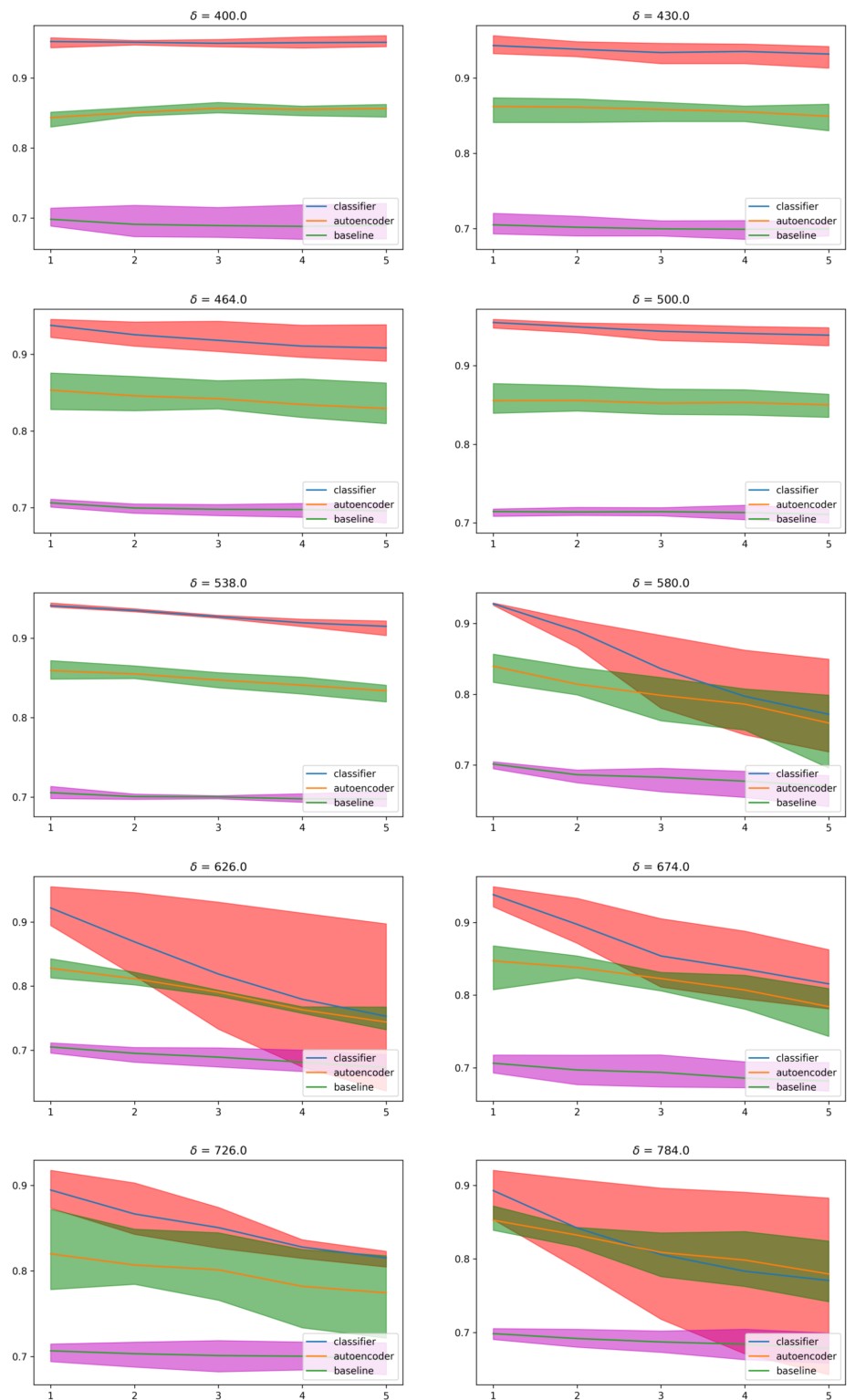

Figure 7: Classification Accuracy on Test Set, i.e., unpermuted MNIST vs. Epochs

[1 Epoch = 1024 (Batch Size) $\times$ 100 (Steps Per Epoch) samples]

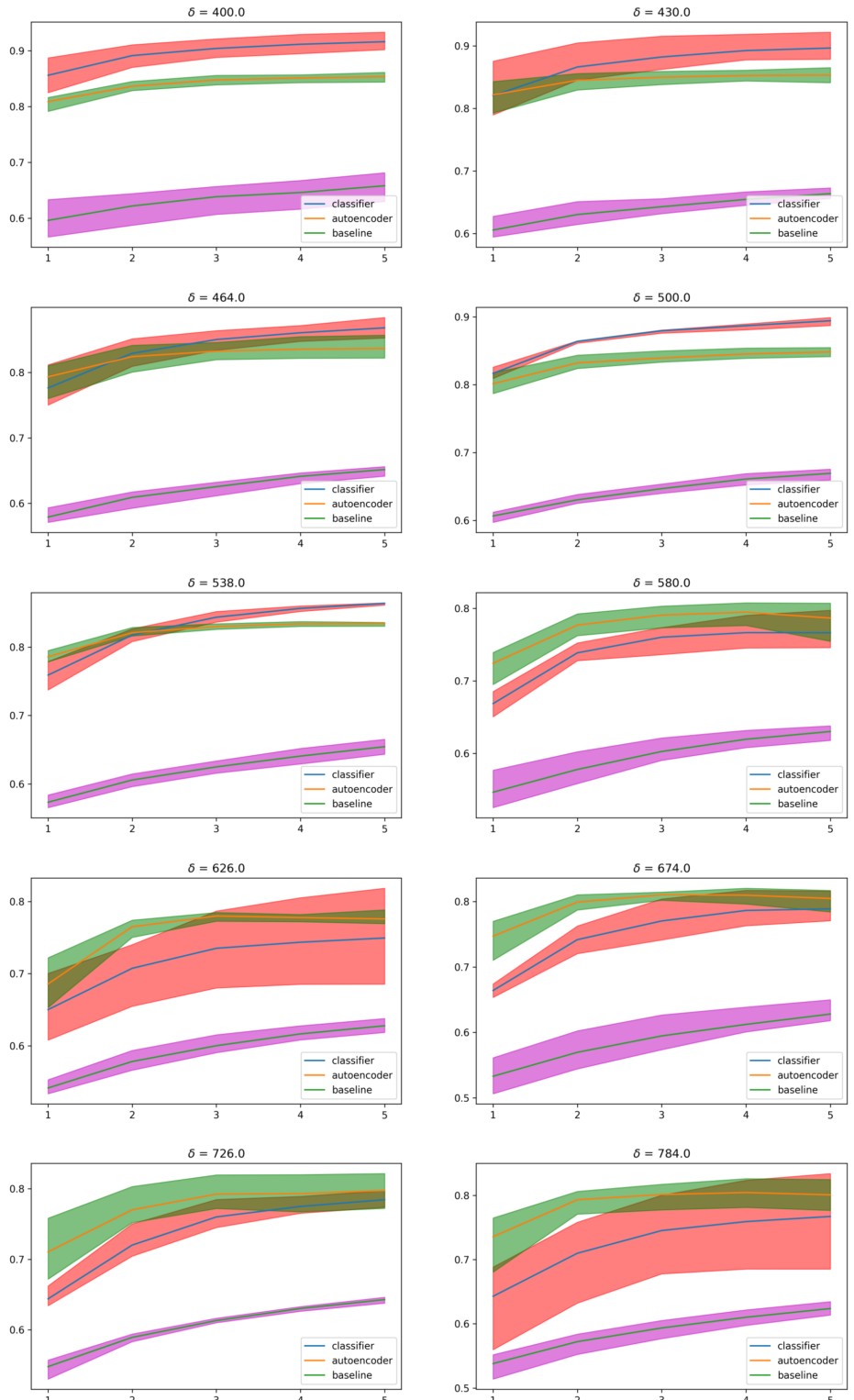

Figure 8: Average Classification Accuracy on Training and Test Sets vs. Epochs

[1 Epoch = 1024 (Batch Size) × 100 (Steps Per Epoch) samples]

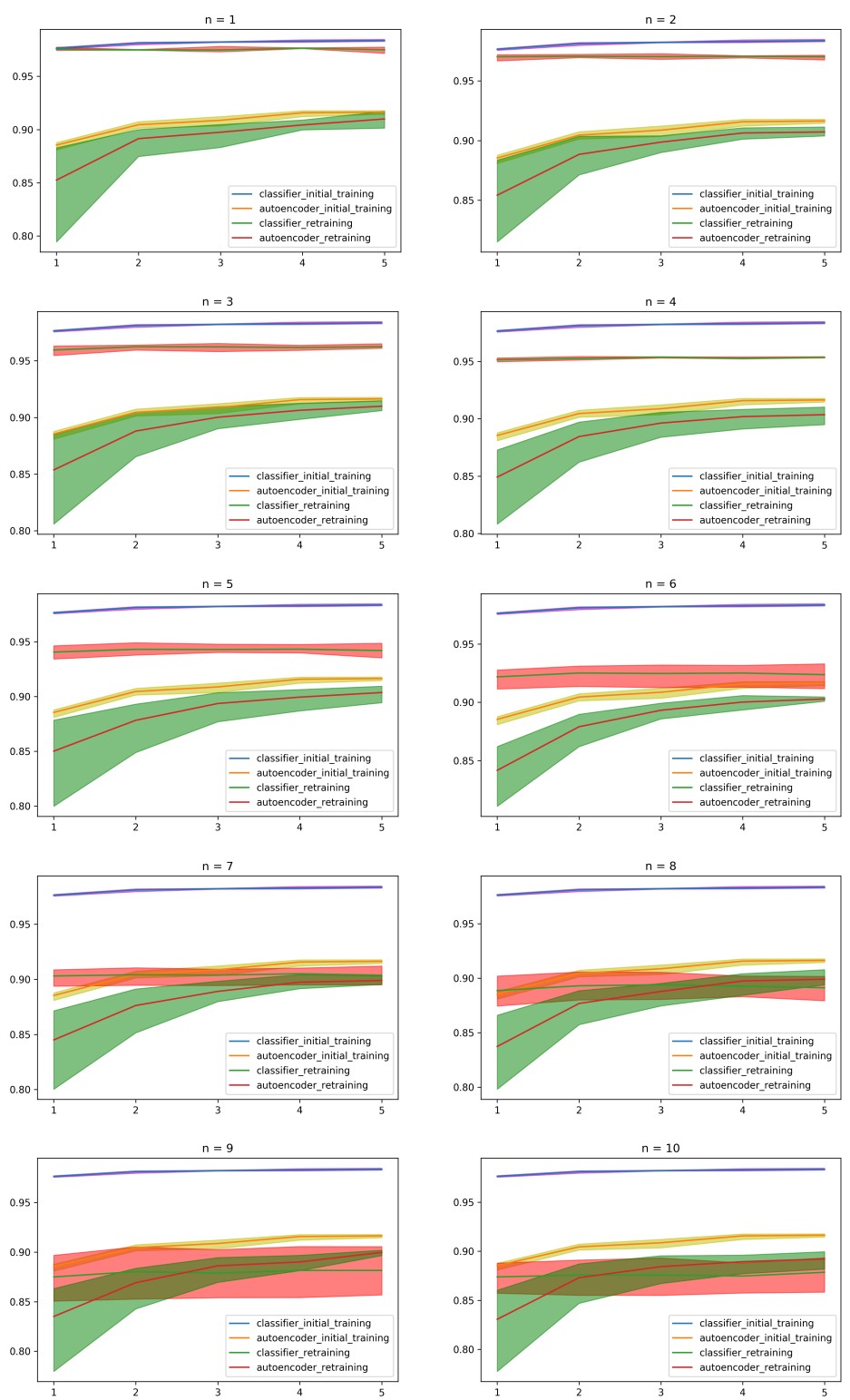

Figure 9: Classification Accuracy on unpermuted MNIST during initial training and retraining after training on $n \in \{1, ..., 10\}$ other permuted MNIST tasks with successively increasing $\delta$'s from those used before vs. Epochs [1 Epoch = 1024 (Batch Size) $\times$ 100 (Steps Per Epoch) samples]

