# OpenReview forum: "Interactions between Representation Learning and Supervision"
_ICLR.cc/2019/Workshop/LLD — Submitted to LLD 2019_

### Official Review · AnonReviewer1 · 2019-04-08
**Review of "Interactions between Representation Learning and Supervision"**

**Rating:** 1
**Confidence:** 2

**Review:**

Summary of the paper:

This paper tries to propose an empirical investigation of the interference/side effect an intermediate representation learning step might have on the supervised learning process.


Reviewer’s assessment:
Besides the fact that the problem is not exposed clearly (even in the introduction) it is particularly hard to make sense of the “Proposed Approach”, and in general to make sense of the whole point of the paper.  The paper is fairly hard to read and is filled with statements having no connection with the previous claims.
Last, for such type of work, a broader diversity of numerical experiments is usually expected.
Hence, I cannot  recommend to accept this paper.

---

### Official Review · AnonReviewer2 · 2019-04-08
**Initial empirical evaluation on the robustness of learned representation to distribution shift**

**Rating:** 3
**Confidence:** 2

**Review:**

Summary

In this paper, the authors ask the question on if we lose general representation by implicitly performing representation learning in supervised settings. In order to empirically answer this question, the authors empirically demonstrate the following 2 observations using permuted MNIST dataset:

1. Representation achieved from unsupervised learning can support fine tuning classifiers for different data distribution and achieve equally well performance. However, representation implicitly derived from supervised learning demonstrates degraded performance for fine-tuning classifier on new data distribution.

2. By using representation achieved from unsupervised learning, a classifier can adapt to an old data distribution after being trained on new distributions, more quickly than using representation derived from supervised learning.

Comments:

1. Technical question: In the computer vision literature, it is well recognized representation learned from supervised training (e.g. using imagenet) generalizes well when fine tuning other dataset in image recognition. However, to the best of my knowledge, there is not too much work on transfer representation from unsupervised learning to other recognition tasks. I was wondering if there is a trade-off between the performance of unsupervised representation and and its generality when the target goal is to achieve better performance for downstream tasks.

2. Quality: In order to better demonstrate the investigation, I would suggest evaluate the learned representation on more different tasks. This will help better validate the claim that unsupervised learning can achieve better generally performant representation. Currently the validation focus on synthetically modified distribution. But I think this is fine as initial work for the workshop.

3. Related work: I would suggest citing literatures in NLP and vision on using supervised/unsupervised representation (e.g. word embeddings, and transfer deep feature extractors in image recognition and other tasks.)

---

### Decision · Program_Chairs · 2019-04-16
**Acceptance Decision**

**Decision:**

Reject

**Comment:**

reivewers found no proposed method and the current empirical analysis too narrow